# Bio-Waste Thermal Insulation Panel for Sustainable Building Construction in Steady and Unsteady-State Conditions

**DOI:** 10.3390/ma12122004

**Published:** 2019-06-22

**Authors:** Miloš Pavelek, Tereza Adamová

**Affiliations:** Department of Wood Processing and Biomaterials, Faculty of Forestry and Wood Sciences, Czech University of Life Sciences, Kamýcká 129, 165 00 Prague 6-Suchdol, Czech Republic; adamovat@fld.czu.cz

**Keywords:** energy sustainability, thermal insulation, rapeseed, woodchips, bio-waste, thermal transmittance, un/steady conditions, VOC emission

## Abstract

Apart from being used as an oil stock for bio-fuels production, an annual crop plant Brassica napus, thought to be an agro-waste, and used either as an animal feed, soil fertilizer or biomass for combustion and thermal energy production. Alternatively, as a bio-based and locally bio-sourced cellulosic material, it could be used as a thermal insulation in sustainable building fabrication, likewise woodchips, a bio-waste from the wood industry. In this study, the above-mentioned bio-waste materials’ thermal properties were identified using a sandwich panel from medium density fibreboard (MDF) and wood studs. Premanufactured panels have been filled in with randomly oriented short-cut rapeseed and with short-cut woodchips. A modified guarded hot box method was used to designate steady and un-steady state thermo-physical parameters of such insulation panels. The examined bio-waste materials absorbed thermal fluctuations of the exterior environment and kept the indoor building environment at constant temperature regardless of such fluctuations. The ability of bio-based sandwich panels to store heat energy was found to be similar to mineral wool. Additionally, VOC (volatile organic compound) emissions of tested materials were identified using gas chromatography-mass spectrometry (GC-MS) combined with headspace solid-phase microextraction (HS-SPME) to declare materials’ harmlessness to indoor environmental quality and human wellbeing. In conclusion, bio-based short-cut materials proved to be a viable environmentally friendly and energy efficient alternative to conventionally used thermal insulations.

## 1. Introduction

In the last decade, an increasing all-society interest in green materials, technologies, and services can be witnessed—partly due to natural resources exploitation, partly due to the constantly growing human liability to natural wealth and expanding consciousness of climate change. A significant amount of research is being carried out to help replace raw chemicals coming from oil feedstock with renewable ones, following the aim to cut down the carbon footprint and ecological human imprint on the environment. These chemical agents and their consequently derived compounds are often sourced from biological commodities, coming from industrial plants production [1]. As a consequence of the emerging technologies and sustainability implementation in technology practice, there has been considerable interest in agro-waste sourced from the industrial crops plantation, especially from oily-seeds plants that are a raw material for bio-diesel synthesis [2]. So far, animal feeds, soil fertilizers, and pellets to be incinerated for heat energy production are the main fields of application of the crop-stalks agro-waste, even though botanical fibers are praised for their low density, specific mechanical properties, biodegradability, low carbon footprint, renewability and affordability [3]. Likewise, there have been considerable traditions over generations in using fibers from plant sources in various applications, especially in textiles and affordable housing construction, straw reinforced mud bricks [4], rammed earth [5] and reed roofs [6] being examples. 

A considerable amount of energy launched by the civilized cultures serves, apart from being used as a source in transportation, as heating facilities in constructions (almost 1/3 to 1/2 of the contribution to CO_2_ emissions) [7]. Not only within the operation, but during the procedure of manufacturing, while installing and throughout the final demolition, the energy embodied in the building materials represents the total energy consumption of a building [8]. Low density, high specific heat capacity and low thermal conductivity represented by a coefficient of < 0.1 W m^−1^ K^−1^ is demanded from a material to qualify as a building thermal insulation [9]. So far, wooden products [10], bamboo [11], straw bales [12], and other industrial crops like, for example: sunflower [13], corn cobs [14], hemp [15] and cotton stalks [16], or less common fibers like Ichu [17] have served as a heat building insulation in structures.

Straw bales are perhaps the most explored naturally available and sustainable thermal insulation and construction material. Their thermal conductivity was reviewed during experimental measurements as a function of density and straw-fiber orientation towards the heat flow. Direction specified, perpendicular thermal conductivity was 0.045–0.056 W m^−1^ K^−1^; then, in a parallel direction, it was 0.056–0.08 W m^−1^ K^−1^, showing increasing values depending on the density of a straw bale [18]. Furthermore, specific thermal conductivities of bio-based and bio-sourced materials: bagasse (0.046–0.055 W m^−1^ K^−1^), kenaf (0.034–0.043 W m^−1^ K^−1^), and pineapple-leaf fibers (0.035–0.042 W m^−1^ K^−1^) are not far from those of conventional man-made materials like mineral wool (0.033–0.040 W m^−1^ K^−1^) and expanded polystyrene (0.031–0.038 W m^−1^ K^−1^) [7]. Thermal conductivities demonstrated by some of the unconventional materials were lower than 0.1 W m^−1^ K^−1^; in case banana and polypropylene (PP) commingled yarn, the highest thermal conductivity results (0.157–0.182 W m^−1^ K^−1^) were reported. Due to anatomical structure of the hollow plant fibers, their thermal conductivities and heat capacities are significantly influenced by the sample’s exposure towards heat flow from the source [18].

Sustainable insulation materials are accessible in several forms—bales, composite boards [19], and sandwich panels [20]; some of them are binder-less [21], some of them are glued together with sustainable plasters or polymer adhesives [22], and some of them may be constructed as a load bearing structures. To determine thermal conductivity of homogeneous materials, several experimental techniques are used—for example: steady state hot-plate, transient hot-bridge, hot-disk, and photo-thermal methods [23]. A hot box apparatus usually determines the thermal performance of complex heterogeneous structural elements built-up of diverse materials. It consists of two enclosed chambers (hot and cold) kept at constant temperatures and separated by the specimen (e.g., wooden panel with agro-waste thermal insulation). Due to the temperature gradient, heat flux between the two chambers through the specimen is measured and thermal resistance of the structure is determined [24]. 

The most prominent oily-seeds plant in Europe nowadays is rapeseed (Brassica napus), an annual plant which grows up to one meter in height. The stem consists of fibers 0.7–2 mm long and with a density of 1550 kg m^−3^, containing 40–50% of cellulose, 25–30% of hemicelluloses and 17–21% of lignin as main chemical constituents [25]. Annual worldwide harvest of rapeseeds is about 12.6 millions of tonnes [26]. Simultaneously, another potentially applicable, and nowadays also accessible, bio-based material is wood [27], especially woodchips [28]. The yearly worldwide production of woodchips is 66.9 millions of tonnes [29]. Combining the use of industrial agro-waste from biofuels production with the requirements on energy savings in building industry, a possibility arises for short-cut rapeseed stalks and short-cut woodchips to be dried and used as a thermal envelope insulation in sustainable buildings construction. 

In order to proclaim the advantages of these unconventional bio-waste thermal insulations (such as low priced, the local economy benefits because of the use of local resources, low energy consumption in production and in-site fastening, do-it-yourself affordable housing projects and simple and ecological end of life disposal), the thermal properties of the overall insulation panel (short-cut rapeseed and short-cut woodchips) were measured with a hot box apparatus. Thermal properties like thermal transmittance of the panel were characterized and the response of the structure towards periodic heating/cooling cycles was determined. The panel’s ability to periodically store and dissipate heat was also determined, in order to show the material efficiency in retaining indoor air at constant conditions (despite the fact that the outside temperatures vary). 

Lastly, the study deals with the indoor air quality while using the short-cut rapeseed and short-cut woodchips as a thermal insulation. Volatile organic compound (VOC) emissions from the tested materials were monitored to further negotiate the potential negative effect of VOCs on human health [30]. The GC-MS analysis was carried out and the specific VOCs were listed. To further support the use of bio-based materials and to contribute to society wellbeing in a sustainable future, every work or research that demonstrates that the long-term performance of sustainable materials, especially in comparison with commonly used conventional materials, is valuable.

## 2. Materials and Methods 

### 2.1. Structure of Raw Materials

Scanning optical microscopy using a binocular magnifier was performed to picture the microstructure of raw materials—rapeseed (Brassica napus) and woodchips from coniferous trees (softwood; bark was deselected) bought from a local supplier. Figure 1a shows the structure of a transversal and a longitudinal section and a surface of the rapeseed stem. A specific oval shape of raw rapeseed stem can be seen in the transversal section. 

A1 detail is focused on a rapeseed pith, a2 detail shows the interface between the pith and the bark (stem outer cortex). A variable shape of woodchips is depicted in Figure 1b.

### 2.2. Insulation Panel Structure—Core

For thermal properties’ examinations, materials bought from a local supplier were shredded one-stage in a hammer mill to short-cut shape and placed into a premanufactured panel. Furthermore, sieve analysis was carried out and additional material characteristics, such as moisture content and VOC emission, were measured.

#### 2.2.1. Moisture Content

For moisture content (*u*) determination, 10 samples of short-cut rapeseed and 10 samples of short-cut woodchips were placed in aluminous pan (each containing 100 g of tested material) and dried for 6 h at 105 °C ± 2 °C in a laboratory conditioning chamber. Afterwards, the absolute moisture content was determined as a percent weight difference between as received (*m_w_*) and dry sample (*m_d_*) and therefore calculated following the equation:
(1)$$u\left(\%\right)=\frac{m_{w}-m_{d}}{m_{d}}\times 100$$

The moisture content of short-cut rapeseed was 8.1 ± 1.6% and 6.9 ± 0.5% of short-cut woodchips.

#### 2.2.2. Fraction Distribution—Sieve Analysis

Sieve analysis was performed to determine the tested short-cut rapeseed and short-cut woodchips fraction. Three randomly collected 100 g samples of each material were tested for fraction size. A screening machine with a laboratory metal sieve according to ISO 3310-1 [31] was employed. 

Table 1 reports short-cut rapeseed fractions distribution obtained from sieve analysis. The fraction 0–8 mm was used as a core layer of tested insulating pane.

The sieve analysis was carried out to observe the abundance of individual fraction width-sections. Although being processed (chopped/cut), short-cut rapeseed preserves the airy cellular morphology responsible for thermophysical properties (Figure 1a). Fractions <0.25 may be considered as a powder that has, contrary to short-cuts, rather isotropic thermophysical properties.

Table 2 demonstrates short-cut woodchips’ fraction distribution obtained from sieve analysis. The most abundant fraction was 0.8–1.6 mm. 

#### 2.2.3. VOC Emission Samples 

Dried-up short-cut rapeseed and short-cut woodchips samples as well as a mineral wool sample were placed into headspace vials (Figure 2). The vials were sealed with magnetic vial caps and were stored for 24 h airtight in a desiccator while preserving constant indoor air temperature (23 °C), corresponding with hot chamber steady state testing conditions.

### 2.3. Insulation Panel Structure—Shell

An MDF (medium density fiberboard) envelope and a 150 mm thick bio-based insulation core sandwich panel of external dimensions 1700 × 1700 × 174 mm^3^—length × height × thickness (Figure 3) was manufactured at the Faculty of Forestry and Wood Sciences, City, Prague to be subjected to thermal loading under steady-state and unsteady-state thermal conditions in a modified guarded hot box.

The thermal insulation was loosely laid and randomly oriented—no binder was used. Bottom and upper MDF sealings were used to slightly press the insulation filler. The MDFs from Egger Ltd. (Hradec Králové, Czech Republic) were used because of their homogeneous cross-section over the entire thickness (12 mm), providing uniform thermal conductivity (λ = 0.14 W m^−1^ K^−1^; ρ = 600–650 kg m^−3^). To ensure structural stability of a panel, the inner cavity was reinforced with joists combining pine studs (40 × 40 × 1700 mm^3^) and HDF (high density fibreboard, λ = 0.17 W m^−1^ K^−1^) of 4 mm thickness. The insulation core thickness was given by the joists’ height (150 mm). All wooden elements, MDF and HDF boards were screwed together with 1.5 × 15 mm^2^ and 3.0 × 30 mm^2^ screws.

The bulk density of an insulation material was calculated as a ratio between the total weight of an insulating filler and the volume of a panel cavity. The bulk density of the short-cut rapeseed insulation panel was 110 kg m^−3^ and, in case of short-cut woodchips insulation panel, it was 205 kg m^−3^. The panel of the same construction was filled with a conventionally used mineral wool thermal insulation.

### 2.4. Modified Guarded Hot Box Design

Thermal properties of tested samples in steady-state and unsteady-state conditions were determined in the modified guarded hot box. The guarded hot box method according to EN ISO 8990 [32] was slightly improved and adjusted in order to reduce heat loss through the hot box envelope. The hot box was composed of two chambers. A hot chamber was supplied with the heating system to maintain high temperatures; on the contrary, a cold chamber was supplied with the cooling system to maintain low temperatures. The tested sample was placed in between. 

The difference between a guarded hot box, designed according to the standardized method, and the modified guarded hot box used in this experiment is illustrated in Figure 4. The box was located in a laboratory with controlled temperature and humidity (HVAC). The temperature on the hot side of the experimental box was kept constant at 24 °C, consistent with the ambient temperature. 

This setting leads to the minimization of heat losses that potentially occurs through the hot chamber walls—heat flow φ3 (Figure 4b). This enables a use of the Hot Box constructed as a Calibrated Hot Box according to EN ISO 8990 [32]. No calibration is needed to minimize the system heat losses through the hot chamber walls. The second most significant advantage of this solution is a minimization of a three-dimensional heat transfer through the sample in a flanking area of the metering chamber—heat flow φ2 (Figure 4a) as standardized in the case of the Guarded Hot Box method.

The modified guarded hot box eliminates heat loss through the chamber wall; therefore, one can assume that the energy supply to the hot chamber flows only through the sample. The gap between the sample and the hot box was filled with an additional mineral wool insulation. The entire sample, as well as the hot box wall perimeter, was sealed with an airtight tape, causing the heat loss to be insignificant. The heat loss and airtightness were checked with a thermal camera after 12 h of sample conditioning.

For this study, a seven-day temperature cycle was selected with the temperature setting of Table 3. The preset temperature program was based on real climate conditions recorded at the local meteorological station, Prague–Suchdol, Czech Republic, in winter 2017.

Each sample was tested under identical conditions with room temperature (22–24 °C) in the hot chamber. The ambient temperature of 22–24 °C was maintained by a stable high performance air conditioning unit (deviation 0.2 °C). The temperature in the cold chamber varied between +6 °C and −13 °C. The current state of the art allows the design of the climate chambers to be precisely controlled and programmed with a temperature program.

The modified guarded hot box construction together with the specific sample position is depicted in Figure 5A. At the beginning of the experiment, a hot chamber was opened and the sample was placed in a hot box to separate the chambers. The temperature difference between the chambers determined the heat flow through the metering area of the sample—φ_1_.

There was a heating system with a maximum power of 500 W in the hot chamber which was regulated by a PID (proportional–integral–derivative) panel controller with an additional thermocouple in the hot chamber. A Power Analyser Rohde & Schwarz HMC 8015 (Rohde & Schwarz, München, Germany) measured the heat flow, as an electric power flows straight to the heating system placed inside the hot chamber. The air temperature was measured using the Data Acquisition Base with humidity and temperature sensors. A set of temperature surface sensors was placed on each side of the measured sample according to EN ISO 8990. The temperature sensors’ location is shown in Figure 5C. Air temperatures and humidity were measured on the cold and hot side 200 mm from the specimen surface (Figure 5B). All of the monitored parameters (air temperature, surface temperature and heat flow transmitted through the sample) were transferred to the computer.

### 2.5. Calculation Procedure in the Steady-State Test

The total thermal transmittance Ut (W m^−2^ K^−1^) of the experimental panel was calculated as the ratio of thermal energy φ_1_ (W) transmitted through the sample area A (m^2^) perpendicular to the heat flow. *T_ai_ − T_ae_* was the difference in air temperatures between the hot and cold sides of the sample (in K):(2)$$U_{t}=\frac{\phi_{1}}{A\left(T_{ai}-T_{ae}\right)}$$

The total thermal resistance Rt (m^2^ K W^−1^) of the experimental panel was calculated as the inverted value of the total thermal transmittance. Referring to Figure 4, the heat flow through the sample (φ_1_ in watts) was determined as input power (φ_p_ in watts). Heat flows φ_3_ and φ_4_ in watts have been neglected.

The input heat flow in the hot chamber was calculated from the electric heater power that was powered and controlled by the PID (proportional-integral-derivative) panel controller with an additional thermocouple in the hot chamber. The electrical output from the PID controller was measured using a Rohde & Schwarz HMC 8015 power analyzer. The metering area of the test panel was 1.7 × 1.7 m^2^. The thermal conductivity calculation of the short-cut rapeseed/short-cut woodchips was based on the thermal resistance calculation R (m^2^ K W^−1^):
(3)$$R_{t}=\left(R_{si}+\underset{R}{\underbrace{\frac{d_{MDF}}{\lambda_{MDF}}+\frac{d_{\frac{rapeseed}{woodchips}}}{\lambda_{\frac{rapeseed}{woodchips}}}+\frac{d_{MDF}}{\lambda_{MDF}}}}+R_{se}\right)$$
where *d* (m) represents the thickness of the material (rapeseed/woodchips/MDF), *R_si_* is the internal surface thermal resistance (m^2^ K W^−1^), *R_se_* is the external surface thermal resistance (m^2^ K W^−1^) and λ (W m^−1^ K^−1^) is the thermal conductivity of the material (rapeseed/woodchips/MDF): (4)$$\lambda_{rapeseed/woodchips}=\frac{d_{rapeseed/woodchips}}{\underset{R}{\underbrace{\left(\frac{(T_{si}-T_{se)A}}{\phi_{1}}\right)}}-\frac{2d_{MDF}}{\lambda_{MDF}}}$$
*T_si_* − *T_se_* represents the difference in surface temperatures between the hot and cold side of the sample (K), *A* was the surface of the panel (m^2^) and *φ*_1_ was the heat flow in the sample (W). The thermal conductivity of the MDF boards *λ_MDF_* (0.14 W m^−1^ K^−1^) was given by the manufacturer.

### 2.6. Experiment Design—Unsteady-State Test

All tested panels were subjected to a one-week dynamic thermal loading. The temperature in the cold chamber (*T_ae_*) fluctuated between +6 °C and −13 °C. The hot chamber temperature (*T_ai_*) was continuously maintained at 24 °C and continuously measured (every min) as an indication of the structure’s reaction to temperature changes. The total heat flow was depending on time, as the total energy transmitted through the panel *E_searched_* (Wh), was calculated using *U_searched_* for every min according to the following equation:
(5)$$E_{searched}=U_{searched}\cdot A\cdot \left(T_{ai}-T_{ae}\right)\cdot t$$
where *U_searched_* is the thermal transmittance (W m^−2^ K^−1^) given by the calculation using a solver function, *A* is the surface of the panel (m^2^), *T_ai_* − *T_ae_* is the difference between hot air temperature and cold air temperature of the panel (K), and *t* is the time (h):
(6)$$Q=\overset{n}{\underset{j=1}{\sum }}{\left(E_{searched,j}-E_{experimental,j}\right)}^{2}$$
where *E_experimental_* is the total energy transmitted through the sample (Wh). To define the *Q*, a variable *U_searched_* to minimize the difference between *E_experimental_* and *E_searched_*, the solver function was used. The thermal conductivity of the insulating core was calculated from the knowledge of the total *U_searched_* of the entire sandwich panel (respectively its inverse values of thermal resistance) and the thermal properties of the MDF envelope of the tested panel (Equation (4)). A similar calculation was used in Burrati et al. [33]. The alternative approach used previously by Pavelek et al. [24] and Trgala et al. [34] brought the opportunity to measure the total energy (Wh) transmitted through the sample and to find appropriate *U* value using dynamic conditions. Heat capacity, thermal response to real weather conditions and the influence of water and vapor content can be taken into account more suitably in the testing method compared to current steady-state conditions. More accurate calculations of the total annual heat loss due to transmission can be assured by the use of long-term real climate temperatures collected at 1 min resolution.

### 2.7. Extraction of Volatiles, GC–MS Analysis and Data Processing

Short-cut rapeseed, short-cut woodchips and mineral wool samples were analysed for their volatile content using gas chromatography coupled to mass spectrometry (GC-MS). To avoid instrumental sensitivity changes, samples were measured in one sequence. For volatile organic compound collection, solid-phase microextraction fiber with a divinylbenzen/carboxen/polydimethylsiloxan (DVB/CAR/PDMS 50/30 µm) coating from Supelco (Supelco Inc., Bellefonte, PA, USA) was employed. Vials were incubated for 10 min to increase volatiles emission from the sample and then volatiles were collected onto a fiber stationary phase for the next 10 min, both at 40 °C.

GC-MS was applied for VOC separation and identification. Basic measurements were performed using Quadrupole Shimadzu GC-MS QP2010 SE-Ultra (Kyoto, Japan), applying an SLB-5MS capillary column (30 m, 0.25 mm i.d., 0.25 µm film thickness) from Supelco. The injection was performed at 250 °C, while the transfer line was kept at 280 °C. The temperature program was as follows: 40 °C for 1 min and then with grad 5 °C min^−1^ to 250 °C and held for 2 min. Total run time was 45 min. Helium was used as a carrier gas at a flow rate of 1 ml min^-1^.

In order to not focus only on a few compounds, the mass analyser was operated in a SCAN mode (scan speed 2000 ns, range 30–400 *m*/*z*. Identification of chemical compounds was based on mass spectral similarity with the in-built NIST MS library (NIST, Gaithersburg, MD, USA; 2017 released version). A group of approximately fifteen main volatile chemical compounds was identified through a literature review to be followed in all of the samples for their comparison. A group of key compounds for each material was defined. Reported intensities are areas of unique mass—the specific mass of compounds’ mass spectrum, which were not coeluting with another compounds signal at a signal’s retention time.

## 3. Results and Discussion

### 3.1. Panel U-Value Calculation from Steady-State Conditions

Experimental measurements were carried out after conditioning of each test sample in a closed hot box. Temperatures were measured at least four hours after steady state was reached, i.e., the temperature fluctuations in the range up to 1%. The temperature in the cold chamber—*T_ae_* (°C)—was set to −13 °C and temperature in the hot chamber—*T_ai_* (°C)—was continuously maintained at 24 °C. Detailed parameters from three experimental measurements, including the thermal transmittance and thermal resistance, are given in Table 4. The average thermal transmittance of the whole sandwich insulating panel filled with short-cut rapeseed was 0.308 ± 0.019 W m^−2^ K^−1^ and the average thermal resistance was 3.255 ± 0.217 m^2^ K W^−1^. The average thermal conductivity of short-cut rapeseed determined under steady-state conditions from all three measurements was 0.048 ± 0.003 W m^−1^ K^−1^. 

All of the surface temperatures were constant during all three tests, with a maximum difference of 0.2 °C on the cold side and 0.1 °C difference on the hot side. The hot air temperature fluctuations were approximately ± 0.05 °C and ± 0.2 °C for cold air temperature (Figure 6). The laboratory ambient temperature was monitored continuously at 24 ± 0.5 °C. 

The average thermal transmittance of the whole sandwich insulating panel filled with short-cut woodchips was 0.403 ± 0.010 W m^−2^ K^−1^ and the average thermal resistance was 2.484 ± 0.060 m^2^ K W^−1^. The average thermal conductivity of short-cut woodchips determined under steady-state conditions from all three measurements was 0.065 ± 0.002 W m^−1^ K^−1^. Detailed parameters from three experimental measurements are given in Table 5.

To compare the results of bio-waste insulation panels, a sandwich panel with mineral wool was measured. The results of the sandwich panel with mineral wool are summarized in Table 6. The average thermal transmittance of the whole sandwich panel was 0.255 ± 0.016 W m^−2^ K^−1^ and the average thermal resistance was determined as 3.930 ± 0.250 m^2^·K·W^−1^. The average thermal conductivity of mineral wool was 0.040 ± 0.003 W m^−1^ K^−1^

### 3.2. Panel U-Value Calculation from Unsteady-State Conditions

The unsteady-state test was performed by changing the air temperature *T_ae_* (°C) in the cold chamber after a certain amount of time, shown in Table 3. The air temperature was controlled by an electronic controller with a preset temperature program. The temperature in the hot chamber *T_ai_* (°C) was continuously set at 24 °C and the temperature in the cold chamber *T_ae_* (°C) was periodically changed between −13 °C and +6 °C for seven days. The preset temperature program was based on average real climate conditions recorded at the local meteorological station, Prague–Suchdol, Czech Republic, in winter 2017. The test panel was always conditioned for 24 h under the pre-set conditions with the air temperature in the hot chamber of 24 °C and air temperature in the cold chamber of 6 °C.

The following data were collected during the experimental tests: the air temperatures *T_ae_* (°C) and *T_ai_* (°C), the surface temperatures *T_se_* (°C) and *T_si_* (°C), and total energy transmitted through the panel (known as an *E_xperimental_* (Wh)) in Figure 7 at time interval of 1 min. The air temperature in the hot chamber *T_ai_* (°C) was continuously recorded every 1 min as an indication of the test panel to thermal impulse. Figure 7 shows the automatic defrosting cycle of the cooling system after approximately 12 h. Total energy (W) transmitted through the test panel was recorded experimentally as a function of time. Furthermore, it was also calculated from Equation (5) and is shown as an *E_searched_* in Figure 7. Figure 7 shows results from two independent measurements, comparing the short-cut rapeseed, the short-cut woodchips and the mineral-wool panel reaction to dynamic thermal conditions as described in the previous paragraph. U_searched_ was found after the experimental measurement using a solver function in MS Excel (version 2016, Microsoft, Redmond, WA, USA).

The steady-state measurements gave more optimistic thermal properties for the panel filled with mineral wool. 

The total thermal transmittance of the entire sandwich panel filled with short-cut rapeseed insulation was 0.271 W m^−2^ K^−1^ and the total thermal resistance value was 3.690 m^2^ K W^−1^. The thermal conductivity of the short-cut rapeseed itself was 0.042 W m^−1^ K^−1^. The total thermal transmittance of short-cut woodchips insulation panel was 0.404 W m^−2^ K^−1^ and the total thermal resistance value was 2.475 m^2^ K W^−1^. The thermal conductivity of the short-cut woodchips was 0.065 W m^−1^ K^−1^.

To compare the results of the bio-waste insulation panels, a sandwich panel with mineral wool was measured. The total thermal transmittance of the entire sandwich panel filled with mineral wool insulation was 0.267 W m^−2^ K^−1^ and the total thermal resistance value was 3.745 m^2^ K W^−1^. The thermal conductivity of the mineral wool was 0.042 W m^−1^ K^−1^. 

Linear thermal transmittance ψ across the reinforcing stud (wood/HDF) was included in the resulting values of the thermal transmittance, thermal resistance, and thermal conductivity.

As can be seen, the U-values of the two tested insulation panels were very close from each other, i.e., panel with short-cut rapeseed filler and mineral wool filler. The short-cut woodchips showed inferior thermal properties compared to these panels. It could be caused due to the bulk density being double that of the short-cut rapeseed panel. It would be appropriate to test panels with identical material filler at various bulk densities. 

Data from measurements conducted under unsteady-state conditions provided comparable values for two tested panels (rapeseed/mineral wool). Therefore, test conditions can strongly influence the performance of a panel in the experimental test, resulting in bio-waste thermal insulation being rejected in favor of conventional thermal insulation. 

Thermal lag, the ability of the system to continuously store and dissipate heat energy when subjected to dynamic thermal loads, is a measure of the wall insulation panel efficiency to keep the indoor build environment at a constant temperature. Wooden sandwich panels with rapeseed short-cut shreddings core showed the same long-term thermal behaviour as wooden sandwich panels with mineral wool core, obviously at the benefit of reduced carbon footprint and lower environmental impact.

### 3.3. VOC Emissions

Within the frame of this work, to better understand the potential impact on indoor environment quality, short-cut rapeseed, short-cut woodchips and mineral wool insulation VOCs were analysed and compared. Fourteen key volatile organic compounds including aldehydes, alkanes, and terpenes were collected especially from bio-waste insulations using HS-SPME GC-MS. Target compounds for bio-based materials (Table 7) were selected based on a list published in ISO 16000-6 (Annex A) focused on building products’ VOC emissions in indoor air [35] as the mineral wool proved to behave like an inert material with almost no VOC emissions. In general, short-cut woodchips’ VOC detection proved to be the most abundant, resulting in a higher number of VOCs as well as higher detector responses observed in short-cut woodchips compared to short-cut rapeseed comparing the bio-based insulations—especially in the case of hexanal and alpha-pinene. An aldehyde pentanal together with even more important aldehyde–hexanal (described as “grassy” [36]) that is known as a product of unsaturated fatty acids oxidation [37] were emitted from a short-cut woodchips sample. Both compounds have been identified as causing unpleasant, irritating odors [38]. Compared to rapeseed and to mineral wool, a wide variety of terpenes (such as alpha-pinene, beta-pinene, camphene, 3-carene) were also observed emitting from woodchips’ samples. 

The chemical composition of raw material used for building insulation materials production represents only one of the factors affecting the quality of indoor air. In addition, the performance of building materials, and therefore VOCs’ release, depends on prevailing thermal and moisture conditions, air pressure difference over the structure, and the quality of construction work [39] (these factors were not considered in presented study). However, it also depends on a structural design with a special focus on wall panel composition of all used materials. Therefore, it is necessary to identify the VOC emissions from insulation materials inside the sandwich panel, determine their environmental impact and select the appropriate enveloping material to prevent their emissions into the interior. It was also addressed in the studies of Little et al. [40], Yuan et al. [41], and Hodgson et al. [42].

## 4. Conclusions

The present study demonstrates the applicability and viability of lignocellulosic bio-waste as a sustainable thermal insulation alternative to mineral-wool. After one-week of dynamic thermal loading, the mineral wool showed a 5% higher U-value compared to steady-state conditions, while short-cut rapeseed showed that a 12% lower U-value and short-cut woodchips reached 0.3% higher U-value. Compared to straw bales, the thermal conductivity and the heat capacity of the insulated wall remained homogeneous across the metering area, providing low thermal energy losses. Moreover, a modified guarded hot box method was successfully used for realistic simulations of thermal behavior of building envelopes and for the determination of insulation panels’ thermal transmittance (U-values) under steady and unsteady state thermal conditions.

From the human wellbeing point of view, lignocellulosic bio-waste materials, especially rapeseed, emitted a similar amount of VOCs comparing to mineral wool. Therefore, it can be considered quite harmless for interiors’ occupants. 

Furthermore, the bio-based sandwich panel allows for complicated structural shapes (elements) engineering. It can be installed easily with loose-fill insulation, giving the architects more freedom in designing sustainable building envelopes, using only a 1.5% thicker layer of rapeseed than mineral wool, while reaching the same thermal resistance.

## Figures and Tables

**Figure 1 materials-12-02004-f001:**
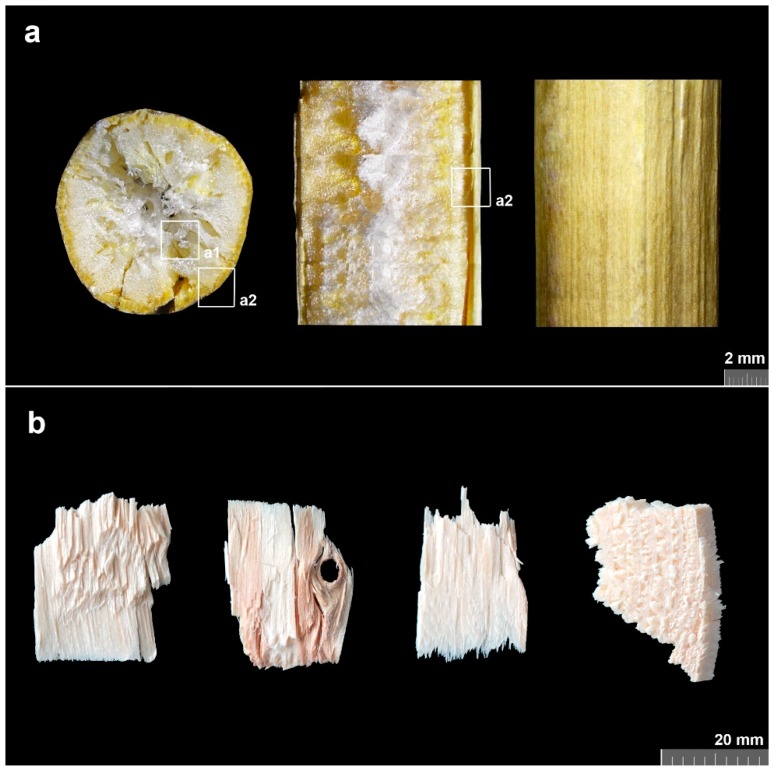
Images of raw materials’ structure from binocular magnifier using scanning optical microscopy: (**a**) rapeseed (Brassica napus) stem; **a1**—rapeseed pith, **a2**—pith and bark interface; (**b**) variability of woodchips shapes.

**Figure 2 materials-12-02004-f002:**
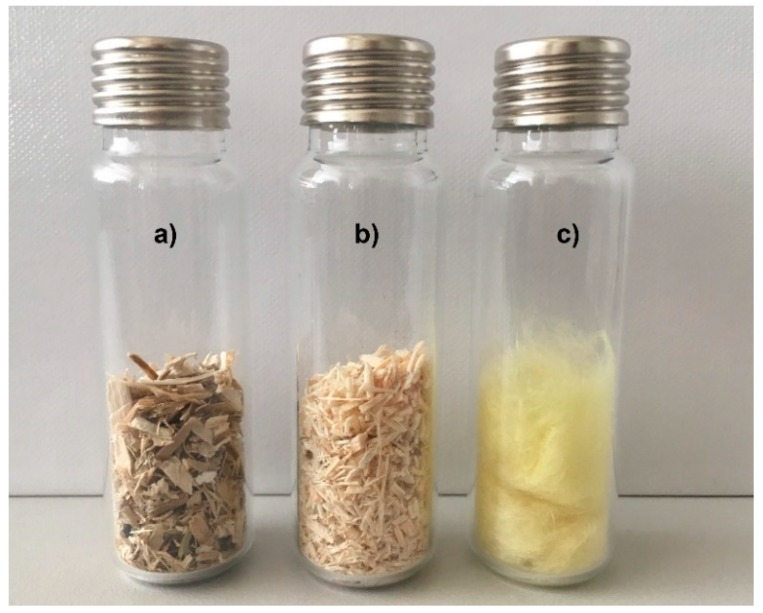
Samples closed in vials ready for GC-MS analysis: (**a**) short-cut rapeseed; (**b**) short-cut woodchips; (**c**) mineral wool.

**Figure 3 materials-12-02004-f003:**
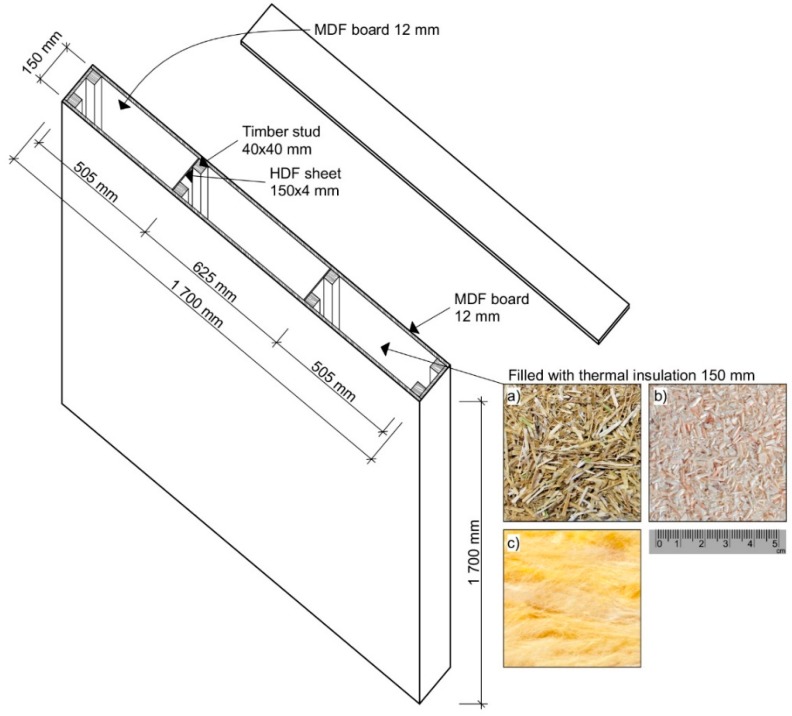
Structure of the insulating sandwich panel (external panel dimensions 1700 × 1700 × 174 mm) filled with: (**a**) short-cut rapeseed; (**b**) short-cut woodchips; (**c**) mineral wool.

**Figure 4 materials-12-02004-f004:**
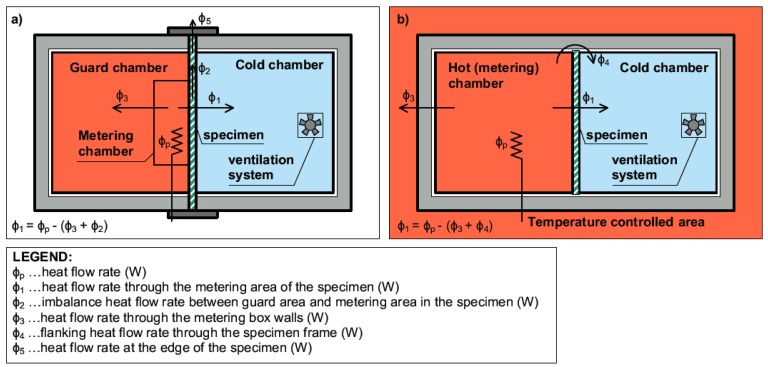
Scheme of the difference between (**a**) a Guarded Hot Box (according to EN ISO 8990 [32]) and the Modified Guarded Hot Box used in this study (**b**).

**Figure 5 materials-12-02004-f005:**
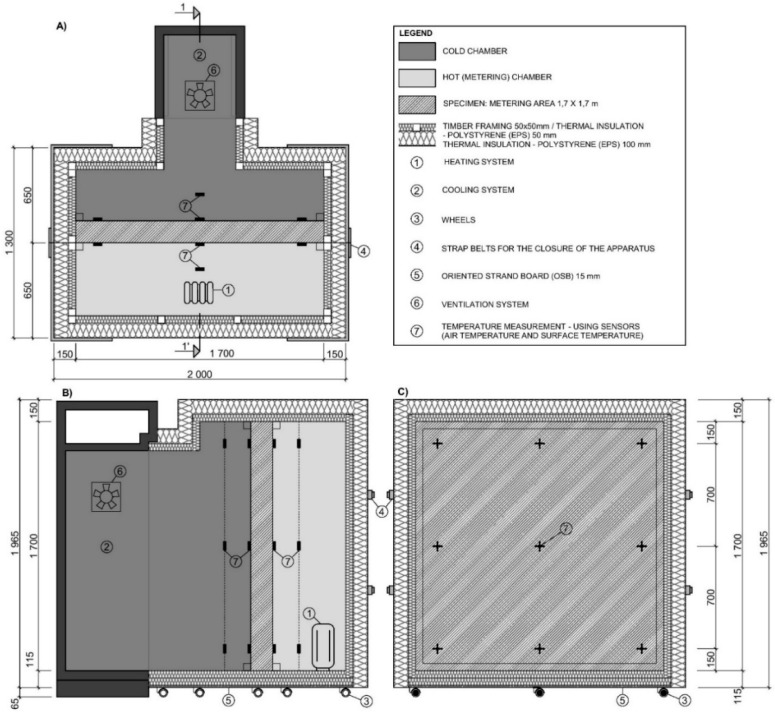
Modified guarded hot box design: (**A**) hot box horizontal section; (**B**) hot box vertical section 1-1’; (**C**) frontal view from the hot/cold side.

**Figure 6 materials-12-02004-f006:**
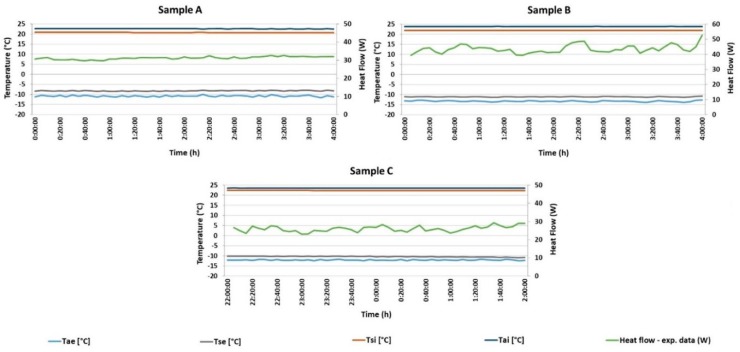
Heat flow and temperature measurements using the steady-state method: (**A**) short-cut rapeseed (test A2 in Table 4); (**B**) short-cut woodchips (test B2 in Table 5); (**C**) mineral wool (test C1 in Table 6); *T_ae_* = air temperature in the cold chamber, *T_ai_* = air temperature in the hot chamber, *T_se_* = surface temperature of sample in the cold chamber, and *T_si_* = surface temperature of sample in the hot chamber.

**Figure 7 materials-12-02004-f007:**
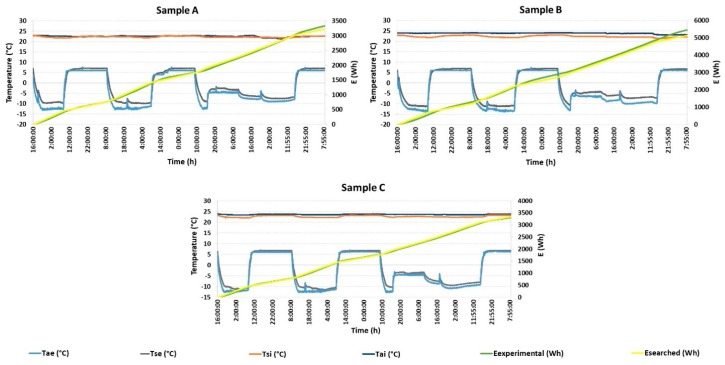
Total energy transmitted through the sample using searched U-value under unsteady-state conditions; (**A**) short-cut rapeseed; (**B**) short-cut woodchips; (**C**) mineral wool; *T_ae_* = air temperature in the cold chamber, *T_ai_* = air temperature in the hot chamber, *T_se_* = surface temperature of the sample in the cold chamber, *T_si_* = surface temperature of the sample in the hot chamber, Eexperimental = total energy transmitted through the sample (Wh), and Esearched = calculated total energy consumption using U-value given by Solver (Wh).

**Table 1 materials-12-02004-t001:** Short-cut rapeseed fraction sizes from a sieve analysis.

Fraction (mm)	<0.25	0.25–0.5	0.5–0.8	0.8–1.6	1.6–2	2–3.15	3.15–8
Fraction representation (%)	1.2	2.8	4.8	39.4	20.1	23.1	8.6

**Table 2 materials-12-02004-t002:** Fractions of short-cut woodchips obtained from sieve analysis.

Fraction (mm)	<0.25	0.25–0.5	0.5–0.8	0.8–1.6	1.6–2	2–3.15	3.15–8
Fraction representation (%)	4.5	7.7	10.3	91.9	7.0	1.1	0

**Table 3 materials-12-02004-t003:** Description of temperature changes in the cold chamber of the Hot Box during the unsteady-state test.

Day	0	1	2	3	4	5	6	7	0	1
**Time**	16:00	16:00	9:00	9:00	9:00	9:00	16:00	9:00	16:00	Repeat
**Temperature**	+6 °C	−13 °C	+ 6 °C	−13 °C	+6 °C	−13 °C	−6 °C	−13 °C	+6 °C	Repeat
**Hours**	24	17	24	24	24	7	17	31	24	Repeat

**Table 4 materials-12-02004-t004:** Thermal transmittance and thermal resistance of sandwich panel filled with short-cut rapeseed insulation under steady-state conditions.

Test	Time (h)	φ_1_ (W)	*d* (m)	*A* (m^2^)	*T_ae_* (°C)	*T_se_* (°C)	*T_si_* (°C)	*T_ai_* (°C)	*R* (m^2^ K W^−1^)	*U* (W m^−2^ K^−1^)
A1	4 h	27.81	0.174	2.89	−11.64	−8.21	21.03	22.64	3.562	0.280
A2	4 h	31.23	0.174	2.89	−10.84	−8.23	20.77	22.58	3.093	0.323
A3	4 h	30.46	0.174	2.89	−11.05	−8.26	20.07	21.74	3.111	0.321

**Table 5 materials-12-02004-t005:** Thermal transmittance and thermal resistance of sandwich panel filled with short-cut woodchips insulation under steady-state conditions.

Test	Time (h)	φ_1_ (W)	*d* (m)	*A* (m^2^)	*T_ae_* (°C)	*T_se_* (°C)	*T_si_* (°C)	*T_ai_* (°C)	*R* (m^2^ K W^−1^)	*U* (W m^−2^ K^−1^)
B1	4 h	42.10	0.174	2.89	−13.41	−11.11	21.94	23.89	2.561	0.391
B2	4 h	43.44	0.174	2.89	−13.31	−11.11	21.92	23.89	2.475	0.404
B3	4 h	44.61	0.174	2.89	−13.39	−11.02	21.88	23.89	2.415	0.414

**Table 6 materials-12-02004-t006:** Thermal transmittance and thermal resistance values of sandwich panel filled with mineral wool insulation under steady-state conditions.

Test	Time (h)	φ_1_ (W)	*d* (m)	*A* (m^2^)	*T_ae_* (°C)	*T_se_* (°C)	*T_si_* (°C)	*T_ai_* (°C)	*R* (m^2^ K W^−1^)	*U* (W m^−2^ K^−1^)
C1	4 h	25.85	0.174	2.89	−12.10	−10.40	22.37	23.53	3.983	0.251
C2	4 h	24.40	0.174	2.89	−11.92	−10.99	22.38	23.60	4.207	0.238
C3	4 h	28.58	0.174	2.89	−12.12	−11.58	22.28	23.49	3.601	0.278

**Table 7 materials-12-02004-t007:** Detected VOCs from specific thermal insulation materials—intensities of monitored VOC emissions; intensities detector response for unique mass.

Insulation Sample	Short-Cut Rapeseed (Brassica Napus)	Short-Cut Woodchips	Mineral Wool
**COMPOUND**	–	–	–
Pentanal	–	30	–
Hexanal	63	180	–
Heptanal	–	11	–
Octanal	–	15	–
Nonanal	14	25	–
alpha-Pinene	9	155	–
beta-Pinene	–	69	–
3-Carene	–	9	–
Decane, 3,6-dimethyl-	–	14	–
Dodecane	12	–	–
Pentadecane	–	–	11
2-t-Butyl-4-methyl-5-oxo-[1,3]dioxolane-4-carboxylic acid	7	–	–
Hydrazine, (1,1-dimethylethyl)-	–	–	66
Nonane, 5-(2-methylpropyl)-	–	–	14
*Note:* selected mass areas were divided by 10^4^		–	–

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
