# Peer review of "Bio-Waste Thermal Insulation Panel for Sustainable Building Construction in Steady and Unsteady-State Conditions"

_materials, 2019, doi:10.3390/ma12122004_

Reviewer 1 Report

See Word file for comments/corrections.

Author Response

Prague, 17. 6. 2019

Dear Reviewer 1,

I would like to express our thanks for a revision of our manuscript – ID materials-530724. We have read it carefully and incorporated most of your comments/suggestions into the updated version. All our responses/scientific opinions are inserted (in red) into the attachment. We very much hope that the current version would be acceptable for publishing in MDPI – Materials.

With compliments, on behalf of authors,

Miloš Pavelek

Reviewer 2 Report

The paper presents a bio-waste thermal insulation panel to sustainable building construction. Authors write that crop plant Brassica can be used as animal feed, soil fertilizer or biomass to combustion and thermal energy production. The paper shows thermal properties which were identified using sandwich panel from medium density fiber-board and wood studs. Pre-manufactured panels were filled with rapeseed and woodchips. Modified guarded hot box method was used to present thermo-physical parameters. Authors present conclusions which indicate that tested bio-waste materials have absorbed thermal fluctuations of exterior environment and kept indoor building environment. Summarizing, they say that bio-based short-cut materials proved environmentally friendly conditions and energy efficient which can be alternative to conventionally used thermal insulations.

Paper looks interesting. The commercial utility of bio-waste as thermal insulation is very value and important topic of science. I can say, that the topic is important and timely.

Introduction chapter is well organized. Used literature positions are correct. Authors made study of thermal conductivity of various materials, and made their comparison. It is good introduction for presented topic. They focused on rapeseeds, what is very popular seed in Europe and North America. They also indicated the problem, especially environmental problem, according the insulation material production.

Chapter 2 presents used materials and applied methods. They explained insulation panel structure of wood and rapeseeds, presenting moisture, fraction distribution analyze, emission problem, insulation panel structure, modification hot box. Authors showed calculation test results according thermal transmittance, based on thermal conductivity. They described experimental test design.

The key chapter – 3 – presents obtained results. Authors showed panel calculation, and emission results.

Last chapter shows conclusions. authors say, that bio-waste insulation panels were manufactured from medium-density fiberboards reinforced with wooden studs and filled with bio-waste rapeseed and woodchips. They conclude that there are no thermal bridges as in case of bale-to-bale connections. Thermal conductivity and heat capacity of obtained wall remains homogeneous across metering area.

Summarizing, I think, the paper is interesting and ready to be published in present form.         

Author Response

Prague, 17. 6. 2019

Dear Reviewer 2,

I would like to express our thanks for a revision of our manuscript – ID materials-530724. Thank you sincerely for your time spent in reviewing this paper and for your additional comments and résumé. We fully appreciate it.

With compliments, on behalf of authors,

Miloš Pavelek

Reviewer 3 Report

The aim of this paper is within the scope of the journal and the written English is clear. Results of thermal transmittance and VOC emission of different materials under investigation are presented. Results are absolutely interesting also for practical application

In  my opinion some aspect are not well clear and more clarification are needed about methodology.

The U value calculation methodology from unsteady state conditions is not well supported by literature (only a paper from the same authors). The Thermal transmittance is by definition a thermal property of a system in steady state condition. The authors seem to find an "equivalent thermal transmission" under a specific variable temperature profile and not directly comparable to steady state methods. In my opinion is not correct to estimate the thermal conductivity from this kind of methodology.

Concerning VOC analysis, I also suggest comments (if of competence) on what would be effects after incapsulation of materials in the MDF sadwich panel

Author Response

Prague, 17. 6. 2019

Dear Reviewer 3,

I would like to express our thanks for a revision of our manuscript – ID materials-530724. We have read it carefully and incorporated your suggestions into the updated version. All our responses/scientific opinions are inserted (in red) into the attachment. We very much hope that the current version would be acceptable for publishing in MDPI – Materials.

With compliments, on behalf of authors,

Miloš Pavelek
